# LibAMM: Empirical Insights into Approximate Computing for Accelerating Matrix Multiplication

**Xianzhi Zeng**[1,2*]    **Wenchao Jiang**[3]    **Shuhao Zhang**[1✉]

[1] National Engineering Research Center for Big DataTechnology and System
Services Computing Technology and System Lab
Cluster and Grid Computing Lab
School of Computer Science and Technology
Huazhong University of Science and Technology, Wuhan, 430074, China
[2] Nanyang Technological University
[3] Singapore University of Technology and Design
shuhao_zhang@hust.edu.cn

## Abstract

Matrix multiplication (MM) is pivotal in fields from deep learning to scientific computing, driving the quest for improved computational efficiency. Accelerating MM encompasses strategies like complexity reduction, parallel and distributed computing, hardware acceleration, and approximate computing techniques, namely AMM algorithms. Amidst growing concerns over the resource demands of large language models (LLMs), AMM has garnered renewed focus. However, understanding the nuances that govern AMM's effectiveness remains incomplete. This study delves into AMM by examining algorithmic strategies, operational specifics, dataset characteristics, and their application in real-world tasks. Through comprehensive testing across diverse datasets and scenarios, we analyze how these factors affect AMM's performance, uncovering that the selection of AMM approaches significantly influences the balance between efficiency and accuracy, with factors like memory access playing a pivotal role. Additionally, dataset attributes are shown to be vital for the success of AMM in applications. Our results advocate for tailored algorithmic approaches and careful strategy selection to enhance AMM's effectiveness. To aid in the practical application and ongoing research of AMM, we introduce *LibAMM* —a toolkit offering a wide range of AMM algorithms, benchmarks, and tools for experiment management. *LibAMM* aims to facilitate research and application in AMM, guiding future developments towards more adaptive and context-aware computational solutions.

## 1 Introduction

Matrix multiplication (MM) is essential across computational domains, particularly in machine learning and scientific simulations. While efforts to improve MM's performance and scalability [24, 38] underscore its importance, MM can dominate up to 90% of processing time in some applications, posing a significant bottleneck. Approximate matrix multiplication (AMM) [17, 12, 34] offers a solution by trading exact accuracy for increased efficiency in contexts where absolute precision is not critical, such as machine learning inference from VGG-like models [6] to GPT-3 LLMs [12]. One may further combine AMM with advances in algorithmic design [32], parallel and distributed computing [4, 26, 37], and hardware technology [11, 18] to mitigate the computational demands of traditional MM, highlighting its potential to revolutionize MM-intensive applications.

---

[*]Work is done while visiting Shuhao Zhang's Lab.

38th Conference on Neural Information Processing Systems (NeurIPS 2024) Track on Datasets and Benchmarks.

The advancement of AMM methods has brought forth a range of techniques designed to meet various computational challenges. These techniques strategically utilize approximations to enhance matrix operation efficiency [13, 16]. Central to AMM is the concept of simplifying computations by modulating calculation detail, prioritizing matrix elements vital to the outcome and downplaying lesser ones [34, 2]. This approach embodies a crucial trade-off: it lowers computational load in exchange for reduced accuracy, a compromise finely adjusted through parameter $\omega$. AMM encompasses three principal approximation strategies: pruning-based methods that cut superfluous calculations for better efficiency [21, 13, 17], extraction-based techniques that identify and leverage key elements or patterns to streamline computations [36, 29, 2, 23, 6, 25], and hybrid approaches that merge these methods [22, 34, 9, 31].

Despite the progress in AMM, a comprehensive and impartial comparison remains elusive, often leading to confusing and sometimes contradictory advice for algorithm selection and task-specific modifications. This confusion stems from several factors: First, the appeal of AMM across various downstream tasks, each favoring a different optimal approach, results in varied conclusions. For example, Adelman et al. [1] favor pruning-based AMM for machine learning training, whereas Blalock et al. [6] recommend extraction-based AMM for inference tasks. Second, the evaluation of some AMM algorithms lacks breadth over representative workloads. Mroueh et al.'s [25] assertion that extraction-based AMM yields only minor inaccuracies is primarily based on synthetic datasets, which may not reflect real-world distribution complexities. Third, inconsistencies in AMM implementation and the standards for baselines raise further issues. The mixture of just-in-time (JIT) and static compilation methods [1, 6, 29] complicates fair performance evaluations among AMM implementations. Additionally, the use of manually coded nested loop MM as a baseline by some studies [36, 25, 16] diverges from modern data science practices, which often employ optimizations like cache-aware data loading and SIMD instructions [28].

To address these gaps, our study delves into a comprehensive empirical analysis of AMM techniques across key algorithmic dimensions. We structure our investigation around three main axes: (a) We conduct a unified static compilation analysis of twelve AMM algorithms alongside two benchmark MM baselines to measure their performance and efficiency. (b) We leverage eight real-world datasets from a spectrum of disciplines to test the versatility and robustness of AMM strategies. (c) We examine four statistics and machine learning applications to assess AMM's practical utility in various downstream tasks. Our methodology intentionally avoids optimizations that cater to specific algorithms or hardware setups, focusing instead on principles with broad applicability [25, 16]. This approach aims to provide a wide-ranging and insightful examination of the AMM domain, highlighting the diverse factors critical for progress in the field. In our study, we discovered insights crucial for advancing MM acceleration via approximation. Key takeaways from our experiments include:

- Among three approximation strategies, **pruning-based and hybrid** AMM **is notably more beneficial** than extraction-based AMM (Section 3.1).
- For all evaluated AMM, **minimizing memory overhead is the key** to practical performance benefits, especially when further leveraging hardware accelerations like GPUs (Section 3.2).
- For all evaluated AMM, the **accuracy greatly depends on dataset attributes** like value skewness and non-zero distribution, requiring improvements of the error bound (Section 3.3).
- In downstream tasks where approximate computing is allowed, **pruning-based and hybrid** AMM **significantly outperform extraction-based** AMM. Specifically, they are superior in both reducing processing latency and conducting the task-aligned approximation with less error (Section 3.4).

To support and inspire ongoing and future research, we introduce *LibAMM*, a framework within the PyTorch ecosystem. *LibAMM* is offered as an open-source tool at https://github.com/intellistream/LibAMM. It aggregates prevalent AMM algorithms, benchmark datasets, and scripts to easily reproduce our experimental results.

## 2 Preliminary

### 2.1 Problem Formulation

Let $A \in \mathbb{R}^{M \times K}$ and $B \in \mathbb{R}^{K \times N}$ denote two matrices intended for multiplication (denoted as $MM(\cdot)$), aiming to calculate the product $C = MM(A, B) \in \mathbb{R}^{M \times N}$. With AMM, the objective shifts towards computing an approximation $\tilde{C} \in \mathbb{R}^{M \times N}$, which seeks to balance computational efficiency with the fidelity of the approximation to $C$. Efforts in AMM focus on developing an algorithm $AMM(\cdot)$

Table 1: AMM algorithms and MM baselines investigated

| Category | Algorithm Name | Descriptions |
|---|---|---|
| Purning-based AMM | INT8 [21] | Purning 32-bit into 8-bit |
| | CRS [13] | Purning elements by sampling |
| | CS [17] | Purning elements by sketching |
| Extraction-based AMM | CoOFD [36] | Extracting singular value, for entire matrices |
| | BLOCKLRA [29] | Extracting singular value, for blocks |
| | FASTJLT [2] | Extracting JL embeddings |
| | VQ [23] | Extracting KNN centroids |
| | PQ [6] | Similar to VQ, more efficient codebook |
| Hybrid AMM | RIP [22] | Randomized JL embeddings extraction |
| | SMP-PCA [34] | Similar to RIP, scaling values for higher accuracy |
| | WEIGTHEDCR [9] | Extract the weight information during sampling |
| | TUGOFWAR [31] | Extract the median and select the optimal after sketching |
| Baseline MM | NLMM [36] | The manual, brute-force, nested loop implementation of MM |
| | LTMM [28] | LibTorch's optimized implementation of MM |

that takes $A$ and $B$ as inputs and applies approximation techniques to produce $\tilde{C}$ with reduced computational demand. The result of this process is expressed as $\tilde{C} = AMM(A, B) \in \mathbb{R}^{M \times N}$, with each element $\tilde{c}_{ij}$ signifying the approximated value corresponding to the element $c_{ij}$ in the traditional product $C$. The critical performance metrics for AMM are *Processing Latency* ($l$), *AMM Error* ($\epsilon$), and *Approximation Impact Factor* ($\Delta E$). $l$ indicates the time efficiency of AMM, by measuring the time from $A$ and $B$ are presented to $\tilde{C}$ is eventually produced. $\epsilon$, calculated by $\frac{\|\tilde{C}-C\|_F}{\|C\|_F}$, measures the *frobenius normalized accuracy* [6, 34] of AMM to standard MM. The $\Delta E$, calculated through $E_{MM} - E_{AMM}$, contrasts the downstream application error when employing MM ($E_{MM}$) against that when using AMM ($E_{AMM}$). This metric effectively measures the impact of AMM on prediction accuracy within downstream tasks, capturing both the potential benefits and drawbacks.

## 2.2 Existing AMM Revisit

Based on how to handle the matrix information, we broadly classify AMM algorithms into three categories: pruning-based, extraction-based, and hybrid. We summarize representative AMM algorithms and the MM baselines in Table 1, more related work is discussed in Section 4.

### 2.2.1 Pruning-based AMM

Pruning-based AMM emphasizes the selective removal of redundant information from matrices. This approach can be implemented at two levels of granularity: bit-wise pruning and element-wise pruning. Bit-wise pruning involves compressing each matrix element by using fewer binary bits. A notable example is INT8, which quantizes 32-bit floating-point elements into 8-bit signed integers. Conversely, element-wise pruning keeps the entire binary representation for certain elements and completely discards others. Techniques for achieving this include CRS (**c**olumn **r**ow **s**ampling) and count sketching mechanisms used in CS (**c**ount **s**ketch-based AMM). Pruning-based AMM algorithms are lightweight, but also reliant on the nature of the underlying data.

### 2.2.2 Extraction-based AMM

Extraction-based AMM focuses on identifying and utilizing higher-level characteristics from matrix elements. It preserves these characteristics in intermediate structures, which are then used to facilitate AMM computations. This approach is favored because operations on these intermediate structures are significantly faster than traditional MM processes. Various matrix attributes serve to accomplish this goal. Frequent Direction-based methods, such as CoOFD, iteratively isolate significant singular values directly from the input matrices. In contrast, the BLOCKLRA algorithm applies singular value decomposition to separate blocks within the matrices, offering a balance between precision and computational efficiency compared to comprehensive singular value decomposition as in CoOFD. FASTJLT employs a different tactic by extracting Johnson-Lindenstrauss (JL) embedding properties of the matrices through Walsh-Hadamard transformations. Beyond leveraging intermediate matrix structures, incorporating a codebook that catalogues K-nearest neighbor (KNN) centroids of matrix rows or columns also facilitates effective information extraction for AMM. This technique is implemented in VQ (vector quantization) and PQ (product quantization), with PQ typically

achieving faster processing times for larger matrices by optimizing the number of KNN centroids through the Cartesian product of subspaces. While extraction-based methods theoretically maintain higher accuracy by preserving essential information during feature extraction, the computational and memory demands of this extraction process may surpass those of standard `MM`.

### 2.2.3 Hybrid `AMM`

Hybrid `AMM` methods combine pruning and extraction techniques to balance processing latency with accuracy. These methods aim to accelerate feature extraction while allowing for selective information pruning to enhance efficiency. For instance, RIP and SMP-PCA introduce randomness to the Johnson-Lindenstrauss (JL) embeddings extraction process, which is inherently deterministic in approaches like FASTJLT, achieving a speed boost. SMP-PCA distinguishes itself by incorporating an additional value scaling step post-randomly pruned JL transform, which theoretically yields higher accuracy compared to RIP. Other hybrid `AMM` strategies focus on refining element-wise pruning through targeted feature extraction. For example, WEIGTHEDCR leverages extracted weight information for more informed sampling, presenting a significant improvement over the indiscriminate sampling seen in CRS. Similarly, the TUGOFWAR method identifies the optimal sketch from a set of random sketches, prioritizing the retention of information over the more arbitrary selection found in CS. By leveraging the strengths of both pruning and extraction, hybrid `AMM` methods stand to deliver superior performance, striking an optimal balance between speed and precision.

### 2.3 Tuning Knob $\omega$ of `AMM` Algorithms

The tuning parameter $\omega$ plays a crucial role in managing the trade-off between computational efficiency ($l$) and accuracy ($\epsilon$)in `AMM` algorithms. This section elucidates the influence of $\omega$ across a spectrum of `AMM` methodologies, noting that $\omega$ has no bearing on INT8, LTMM, and NLMM. The latter two algorithms, being precise `MM` implementations, and INT8, although an `AMM` algorithm, do not utilize this adjustable parameter.

VQ and PQ showcase an innovative approach by adjusting the number of K-nearest neighbor (KNN) centroids relative to the row count of the input matrix $A$ through $\omega$. Unlike other `AMM` methods that rely on intermediate matrices, these algorithms employ codebooks for feature encapsulation, linking each KNN centroid to a unique code within the codebook. This adjustment effectively moderates the codebook's capacity and computational overhead, striking a balance between latency and precision.

For feature extraction-focused algorithms like CoOFD and FASTJLT, $\omega$ is instrumental in determining the dimensionality of the feature space extracted in relation to input matrix $A$'s column volume. By converting singular values or JL embeddings of $A, B$ into intermediate matrices, the size of which is configurable through $\omega$, these algorithms propose a trade-off: smaller intermediate structures facilitate quicker computations at the expense of accuracy, while larger configurations promise enhanced accuracy at the cost of increased computational time. This principle is similarly applicable to hybrid `AMM` approaches such as RIP and SMP-PCA, which incorporate randomized optimizations in the feature extraction phase to improve efficiency.

In the context of BLOCKLRA, $\omega$ significantly impacts the sizing of the feature extraction matrix in proportion to the eigenspace of $A, B$, diverging from the full matrix consideration to a block-wise feature extraction perspective. This nuanced application of $\omega$ alters the dimensionality of intermediate matrices, thereby modifying computational dynamics.

Lastly, for CRS, CS, WEIGTHEDCR, and TUGOFWAR, the parameter $\omega$ governs the proportion of matrix elements preserved after pruning, employing sampling or sketching techniques to forgo computations on specific elements within $A, B$. The allocation determined by $\omega$ serves to represent the characteristics of the majority pruned, leveraging statistical properties such as mean or variance to approximate the impact of excluded elements.

## 3 Empirical Studies

In the following section, we first introduce experimental configurations and then present the results.

**Evaluation Configurations.** Our experimental framework is meticulously designed to ensure a comprehensive and equitable assessment of `AMM` algorithms across a spectrum of real-world and

| Name | Application Field | Size |
|------|-------------------|------|
| *ECO* | Economics | $207 \times 260$ |
| *DWAVE* | Integrated Circuit | $512 \times 512$ |
| *AST* | Astrophysics | $765 \times 765$ |
| *UTM* | Plasma Physics | $1700 \times 1700$ |
| *RDB* | Chemical Engineering | $2048 \times 2048$ |
| *ZENIOS* | Air Traffic | $2873 \times 2873$ |
| *QCD* | Quantum Physics | $3072 \times 3072$ |
| *BUS* | Land Traffic | $4929 \times 10595$ |

Table 2: Real-world Workloads for Comparing AMM Algorithms

| Downstream Task | Dataset | Matrix Size | Proportion |
|-----------------|---------|-------------|------------|
| PCA [33, 30] | *SIFT10K* | $128 \times 10000$ | 13.2% |
| Machine Learning Training [1] | *MNIST* | $392 \times 60000 \cdot 60000 \times 392$ | 21.4% |
| Machine Learning Inference [6] | *CIFAR100* | $10000 \times 512 \cdot 512 \times 100$ | 86.4% |
| Unitary Transformation [20, 15] | *QCD* | $3072 \times 3072$ | 100% |

Table 3: Dataset and Latency Proportion of AMM-Replaceable MM in Downstream Tasks

synthetic datasets. To this end, we delineate our evaluation methodology under two principal components: the datasets employed and the detailed implementation nuances of AMM algorithms. In the evaluation, we set $\omega$ as $10\%$ if not otherwise specified.

**Datasets.** The core of our benchmark suite is derived from MatrixMarket [27], encapsulating a diverse array of **real-world workloads** including but not limited to *ECO*, *DWAVE*, *AST*, *UTM*, *RDB*, *ZENIOS*, *QCD*, and *BUS*, detailed in Table 2. These datasets are preprocessed to normalize matrix elements within the range of $-1$ to $+1$, covering a wide breadth of applications from economic modeling (*ECO*) to power flow analysis (*BUS*). Specifically, we linearly align the maximum value to 1 and minimum value to -1, and we leave the more complicated and application-specific normalization such as L2 Normalization [28] for future works. Complementing these real-world datasets, we also generate **synthetic workloads** using LibTorch functionalities like *torch::rand*.

**Downstream Tasks.** We examine a suite of downstream tasks where the integration of AMM is particularly appealing. These tasks include *Principal Component Analysis* (PCA) [33, 30], *Machine Learning Training* [1] and *Inference* [6] phases, and *Unitary Transformation* [20, 15]. Detailed descriptions of those applications are presented in Appendix A. Note that, while some pruning-based AMM methods under element quantization like INT8 can seamlessly integrate into complex models, such as transformer-based large language models [35, 12], incorporating other AMM strategies (e.g., CRS or SMP-PCA) into these advanced models poses a significant challenge. As such, we follow [1] for training and [6] for inference, and leave more intricate models in future research.

Table 3 presents an analysis of the datasets utilized, highlighting the proportion of latency associated with AMM-replaceable MM operations within these downstream tasks. By integrating these tasks into the LibTorch ecosystem, we capitalize on its LTMM functionality for MM operations. The machine learning training and inference procedures, in particular, engage the PyTorch frontend. We bind PyTorch calls to static compilation, thus isolating the impacts of JIT on the execution of AMM or MM.

**Implementation.** We unify the implementation of the examined AMM algorithms into one C++ codebase, using static compilation to guarantee consistency across experiments. We use the IEEE 754 32-bit floating-point (FP32) format for representing matrix elements, and take the advantage of LibTorch C++ API [28], thereby inheriting AVX-512 instructions of FP32 from LibTorch. While certain AMM algorithms might benefit from algorithm-specific optimizations—like Bernoulli sampling probabilities in CRS [1] or the *MADDNESS* hash function in PQ [6]—we exclude these from our primary evaluation. This exclusion is to avoid bias introduced by optimizations that rely on assumptions not universally applicable, aiming for evaluations that are as inclusive and applicable as possible. We focus on in-memory MM and AMM, and leave out-of-memory case [38] or disk-memory corporation [3] for future works.

**Deployment.** Our evaluation primarily unfolds on a Silver 4310 processor, with both MM and AMM seamlessly adapting to parallel and distributed computing through a straightforward block partition approach [6]. This directs our focus towards single-threaded evaluation. Notably, we include an experiment to explore AMM's performance in a parallelized context, utilizing an I7-13700K CPU and an RTX A6000 GPU, to assess its potential on parallel hardware architectures.

## 3.1 Algorithmic Strategies

We first investigate how different algorithmic strategies affect the effectiveness of AMM, concerning both processing latency and accuracy, as summarized in Table 4.

| Algorithms | | Processing Latency $l$ ($\times 10^3$ ms) | | | | | | | | AMM Error $\epsilon$ | | | | | | | |
|---|---|---|---|---|---|---|---|---|---|---|---|---|---|---|---|---|---|
| | | ECO | DWAVE | AST | UTM | RDB | ZENIOS | QCD | BUS | ECO | DWAVE | AST | UTM | RDB | ZENIOS | QCD | BUS |
| Pruning-based | INT8 | 0.01 | 0.03 | 0.03 | 1.09 | 0.95 | 2.23 | 2.04 | 13.07 | 0.0193 | 0.0234 | 0.0122 | **0.0200** | 0.0102 | 0.0000 | 0.0001 | 0.0034 |
| | CRS | **0.10** | **0.10** | **0.91** | **0.90** | **0.96** | **0.99** | **1.06** | **1.88** | 0.5939 | 0.0203 | 0.0001 | 2.4437 | 0.0050 | 0.0003 | 0.0044 | 1.0000 |
| | CS | 0.18 | 0.21 | 0.48 | 1.12 | 1.48 | 2.21 | 2.64 | 14.28 | 0.3754 | 0.3015 | 0.0602 | 2.4521 | 0.0287 | 0.0668 | 0.0168 | 0.0034 |
| Extraction-based | CoOFD | 3.09 | 44.72 | 31.73 | 277.81 | 730.73 | 1092.58 | 1302.97 | 7999.6 | 0.1587 | 0.0353 | 0.0131 | 0.9640 | 0.0088 | 0.0038 | 0.0059 | 0.0000 |
| | BlockLRA | 1.17 | 3.72 | 7.78 | 46.70 | 20.23 | 43.53 | 50.40 | 31.45 | 0.1129 | 0.0004 | 0.0000 | 0.8705 | 0.0001 | 0.0000 | 0.0001 | 0.0770 |
| | FastJLT | 2.32 | 2.35 | 6.28 | 29.63 | 31.45 | 40.82 | 40.74 | 621.69 | 0.4654 | 0.2205 | 0.1514 | 2.4824 | 0.0770 | 0.0038 | 0.1177 | 0.1002 |
| | VQ | 2.90 | 4.79 | 7.37 | 31.18 | 51.34 | 112.94 | 136.17 | 1088.6 | 0.2780 | 0.0019 | 0.0000 | 0.9211 | 0.0003 | 0.0000 | 0.0005 | 0.0000 |
| | PQ | **3.96** | 1169.5 | 50.64 | 102.32 | 124.85 | 186.81 | 202.07 | 531.48 | 0.5436 | 0.0000 | 0.0065 | 0.9502 | 0.0039 | 0.0010 | 0.0016 | 0.0001 |
| Hybrid | RIP | 0.75 | 0.91 | 1.26 | 1.60 | 1.80 | 1.60 | 2.08 | 7.15 | 0.4502 | 0.0846 | 0.3370 | 2.4760 | 0.1733 | 0.0060 | 0.0396 | 0.0238 |
| | SMP-PCA | **1.81** | **1.34** | **1.54** | **2.48** | **2.68** | **2.51** | **3.16** | **6.97** | 0.3618 | 0.0029 | 0.0001 | 2.4676 | 0.0004 | 0.0000 | 0.0002 | 0.0031 |
| | WEIGTHEDCR | 0.48 | 0.39 | 1.46 | 1.57 | 1.45 | 1.46 | 1.72 | 2.09 | 4.1807 | 0.0185 | 0.0008 | 2.5278 | 0.0047 | 0.0002 | 0.0044 | 5217.1 |
| | TugOfWar | 12.70 | 25.66 | 27.56 | 25.29 | 25.76 | 42.17 | 42.32 | 55.46 | 0.4421 | 0.0368 | 0.0926 | 2.4753 | 0.0058 | 0.0112 | 0.0377 | 0.0032 |
| Baselines | NLMM | 1.41 | 4.01 | 10.72 | 124.41 | 433.57 | 704.94 | 1418.78 | 9244.8 | 0.0000 | 0.0000 | 0.0000 | 0.0000 | 0.0000 | 0.0000 | 0.0000 | 0.0000 |
| | LTMM | **0.10** | **0.12** | **0.11** | **4.85** | **6.52** | **11.14** | **11.43** | **60.60** | 0.0000 | 0.0000 | 0.0000 | 0.0000 | 0.0000 | 0.0000 | 0.0000 | 0.0000 |

Table 4: Overall performance comparison of processing latency $l$ and AMM Error $\epsilon$.

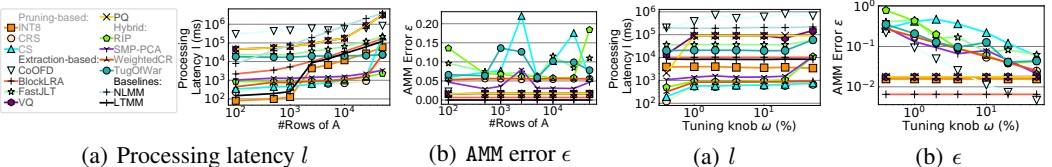

| (a) Processing latency $l$ | (b) AMM error $\epsilon$ | (a) $l$ | (b) $\epsilon$ |
|---|---|---|---|

| Figure 1: Scalability to data volume. | Figure 2: Impacts of tuning knob $\omega$. |
|---|---|

Observation 1. Contrary to common belief [36, 29, 2, 23, 6, 25], pruning-based and hybrid strategies are more practically useful than extraction-based strategy.

**In sufficiently large datasets, pruning-based AMMs, especially CRS and INT8, significantly outperform others,** with CRS reducing latency by 56.3% and 99.9% against the LTMM benchmark in *ECO* and *BUS*, respectively. While extraction-based methods typically show higher latencies than both LTMM and NLMM, PQ within this group stands out, albeit still lagging behind pruning-based solutions. Hybrid AMMs, notably SMP-PCA, offer a compromise, demonstrating substantial efficiency improvements on *BUS* over LTMM.

**Extraction-based AMMs tend to deliver higher accuracy than pruning-based approaches**, illustrated by the near-zero errors in *BUS* dataset scenarios, in stark contrast to the higher errors associated with pruning methods like CRS. Hybrid AMMs, with SMP-PCA as a prime example, strike a balance, achieving low errors in certain datasets and showcasing the trade-off between efficiency and accuracy. This underscores the need for careful selection of AMM strategies to align with the specific accuracy requirements of an application.

Observation 2. Pruning-based AMM faces scalability challenges at larger data sizes.

In our analysis focused on the scalability of AMM algorithms, matrices $A, B$ were generated through the *torch::rand* function, assigning element values between 0 and 1. Keeping the columns of $A$ and the dimensions of $B$ constant at 2500, we varied the row count of $A$ from 100 to 50000. Results are detailed in Figures 1(a) and 1(b).

**A critical takeaway from our findings is that pruning-based methods encounter notable scalability challenges when data sizes increase.** While initially demonstrating latency reductions at moderate data scales (1000 ∼ 10000 rows), algorithms such as INT8, CRS, and CS begin to struggle as the dataset size exceeds certain thresholds. This scalability issue is particularly marked when comparing the latency spike observed in these pruning-based algorithms against the more stable performance of hybrid and extraction-based methods at larger dataset sizes. Hybrid AMM methods, for instance, while showing a balance between efficiency and accuracy, do not experience the same degree of scalability challenges as their pruning-based counterparts. Similarly, extraction-based algorithms, although generally slower, maintain a consistent error rate and do not exhibit the same sharp increase in processing latency with data scale.

Observation 3. There is a complex performance trade-off of AMM due to $\omega$.

In our analysis of the tuning parameter $\omega$'s influence on AMM algorithms, we utilized input matrices $A, B$ with $2500 \times 2500$ dimensions, generated via *torch::rand*. By varying $\omega$ between 0.04% and 50%, we observed its impact on processing latency ($l$) and AMM error ($\epsilon$), with findings illustrated in Figure 2. It's important to note that $\omega$ does not affect LTMM, NLMM, or INT8 (Section 2.3).

| Algorithms | | Smallest Dataset *ECO* (207 × 260) | | | | | | Moderate Dataset *UTM* (1700 × 1700) | | | | | | Largest Dataset *BUS* (4929 × 10595) | | | | | |
|---|---|---|---|---|---|---|---|---|---|---|---|---|---|---|---|---|---|---|---|
| | | Mem Stall | L1D Stall | L2 Stall | L3 Stall | C_Stall | Useful | Mem Stall | L1D Stall | L2 Stall | L3 Stall | C_Stall | Useful | Mem Stall | L1D Stall | L2 Stall | L3 Stall | C_Stall | Useful |
| Pruning-based | INT8 | 99.12 | 0.55 | 0.21 | 0.12 | **0.00** | 0.00 | 73.96 | 24.91 | 1.11 | 0.01 | **0.00** | 0.00 | 54.40 | 29.45 | 9.10 | 7.05 | **0.00** | 0.00 |
| | CRS | 32.34 | 25.07 | 23.23 | 19.36 | **0.00** | 0.00 | 22.21 | 1.73 | 1.59 | 1.35 | **59.30** | 13.82 | 62.44 | 17.60 | 9.99 | 8.70 | **0.00** | 1.27 |
| | CS | 82.72 | 8.61 | 4.77 | 3.90 | **0.00** | 0.00 | 56.76 | 21.85 | 14.12 | 7.27 | **0.00** | 0.00 | 29.05 | 24.62 | 23.93 | 22.39 | **0.00** | 0.00 |
| Extraction-based | CoOFD | 22.27 | 0.59 | 0.40 | 0.26 | **62.37** | 14.11 | 24.10 | 0.21 | 0.07 | 0.06 | **58.52** | 17.03 | 23.53 | 0.57 | 0.44 | 0.25 | **59.45** | 15.76 |
| | BlockLRA | 45.26 | 2.47 | 1.37 | 0.93 | **11.68** | 38.29 | 75.72 | 8.37 | 5.16 | 3.55 | **0.00** | 7.20 | 80.82 | 0.55 | 0.29 | 0.23 | **0.00** | 18.10 |
| | FastJLT | 20.43 | 0.59 | 0.50 | 0.40 | **65.54** | 12.54 | 79.48 | 0.53 | 0.30 | 0.24 | **0.00** | 19.45 | 96.71 | 2.39 | 0.52 | 0.39 | **0.00** | 0.00 |
| | VQ | 21.02 | 1.06 | 0.89 | 0.73 | **63.46** | 12.84 | 33.09 | 23.08 | 22.15 | 21.68 | **0.00** | 0.00 | 27.81 | 24.56 | 23.97 | 23.66 | **0.00** | 0.00 |
| | PQ | 21.62 | 0.66 | 0.55 | 0.42 | **63.03** | 13.72 | 20.71 | 0.73 | 0.59 | 0.49 | **64.68** | 12.80 | 26.13 | 5.43 | 4.91 | 3.37 | **44.53** | 15.63 |
| Hybrid | RIP | 20.65 | 0.76 | 0.68 | 0.47 | **64.80** | 12.63 | 25.60 | 1.89 | 1.14 | 0.84 | **53.26** | 17.27 | 79.62 | 12.91 | 4.25 | 3.23 | **0.00** | 0.00 |
| | SMP-PCA | 20.36 | 0.58 | 0.55 | 0.43 | **65.57** | 12.51 | 24.22 | 1.91 | 1.13 | 0.93 | **55.80** | 16.02 | 65.76 | 16.90 | 9.85 | 7.49 | **0.00** | 0.00 |
| | WeigthedCR | 21.66 | 2.25 | 2.10 | 1.70 | **59.62** | 12.68 | 21.25 | 2.28 | 1.70 | 1.48 | **60.10** | 13.20 | 38.66 | 26.43 | 19.08 | 15.83 | **0.00** | 0.00 |
| | TuGoFWar | 20.26 | 0.34 | 0.30 | 0.19 | **66.41** | 12.50 | 21.79 | 0.73 | 0.46 | 0.33 | **62.86** | 13.83 | 46.15 | 4.97 | 1.80 | 1.32 | **8.19** | 37.57 |
| Baselines | NLMM | 98.55 | 0.88 | 0.33 | 0.23 | **0.00** | 0.00 | 50.23 | 25.33 | 14.36 | 10.09 | **0.00** | 0.00 | 40.78 | 27.24 | 20.20 | 11.77 | **0.00** | 0.00 |
| | LTMM | 39.16 | 23.48 | 21.41 | 15.95 | **0.00** | 0.00 | 98.73 | 0.35 | 0.13 | 0.10 | **0.00** | 0.68 | 98.52 | 1.17 | 0.17 | 0.14 | **0.00** | 0.00 |

Table 5: The proportion of processor cycles (%). *Mem Stall*, *L1D Stall*, *L2 Stall* and *L3 Stall* are disjointly caused by pending memory operations, while *Computing Stall* (C_Stall) is caused by pending computing operations.

**Our results highlighted a complex performance trade-off landscape.** Generally, increasing $\omega$ leads to lower $\epsilon$ for most algorithms, except CS, but at the expense of increased $l$. This trade-off between minimizing error and managing latency varies significantly among algorithms. For instance, CRS can reduce $l$ by up to $60\%$ by tolerating an $\epsilon$ rise from $0.02$ to $0.28$. In contrast, BlockLRA consistently keeps $\epsilon$ below $0.01$ and indicates a narrower trade-off scope. Interestingly, CS deviates from the expected trend of reduced $\epsilon$ with higher $\omega$, attributed to its relatively loose error bound.

## 3.2 Operational Specifics

We examined the impact of operational specifics on AMM performance, categorizing processor operations into *memory operations* and *computing operations*, using PAPI [8] to trace processor cycles during AMM. Our analysis on a Silver 4310 processor identified cycles affected by memory operations as 1) *Mem Stall*, 2) *L1D Stall*, 3) *L2 Stall*, and 4) *L3 Stall*. Additionally, we noted 5) *Computing Stall*, caused by pending computing operations, and 6) *Useful* cycles, where the processor efficiently executes without memory or computing stalls. Importantly, stalls are only recorded when operations exceed their expected completion time [19], with on-time operations classified as *Useful*.

> Observation 4. Cache and memory stalls significantly impact all AMM algorithms, particularly in larger datasets, with memory stalls been a key issue.

In Table 5, we delve into the scalability of AMM algorithms across a spectrum of dataset sizes, including *ECO*, *UTM*, and *BUS*. A key finding is the pronounced impact of cache and memory stalls on all AMM algorithms and MM baselines, particularly in larger datasets, where memory stalls emerge as a critical bottleneck. For instance, FastJLT sees nearly all its processor cycles consumed by memory stalls in the *BUS* dataset, underscoring the heavy data demands of matrix operations.

**Memory access and the resulting cache performance issues scale with dataset size**, evident in the stark increase from $21.91\%$ of affected cycles in *ECO* to $99.99\%$ in *BUS* for SMP-PCA. Memory stalls notably surpass cache stalls in magnitude across most algorithms, such as FastJLT on *BUS*, due to LibTorch's cache optimizations aligning well with algorithms that utilize contiguous data structures. Conversely, algorithms like CS and VQ face greater challenges due to their access patterns to disjoint data structures. Furthermore, computational challenges compound for algorithms like CoOFD and PQ in large datasets, with a significant portion of their processing time hampered by computational stalls—evident in the substantial delays faced by these algorithms on *BUS*. These observations suggest that while memory stalls are a universal bottleneck, especially in larger datasets, computational efficiency remains a critical consideration for certain AMM algorithms.

Figure 3 further reveals **the escalation of memory stall cycles with data size**, aligning with increases in processing latency ($l$) as demonstrated in Figure 1(a). The surge in memory stalls, particularly observed in LTMM with over $71\times$ growth from 1000 to 2500 rows, highlights the critical point of memory bandwidth utilization. Pruning-based and hybrid AMM algorithms, notably SMP-PCA and RIP, excel in mitigating this memory stall growth, outperforming even LTMM by maintaining stall cycles within manageable limits up to 50000 rows. This contrasts with extraction-based algorithms, which exhibit higher memory stall cycles due to their more intensive memory usage.

> Observation 5. Adapting AMM to GPUs shows promise but is limited by data transfer costs, highlighting the need for optimized hardware-software integration.

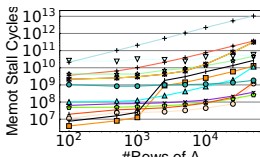
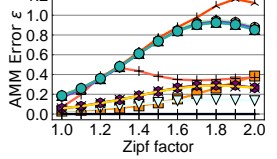
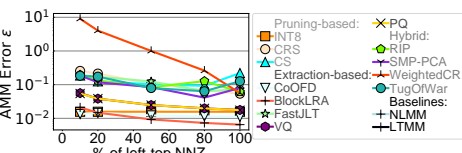

Figure 3: Memory stall with growing data scale.

Figure 4: Impacts of skewed numeric values.

Figure 5: Impacts of biased NNZ region.

| Algorithms | Smallest Dataset *ECO* (207 × 260) | | | Moderate Dataset *UTM* (1700 × 1700) | | | Moderate Dataset *RDB* (2048 × 2048) | | | Largest Dataset *BUS* (4929 × 10595) | | |
|---|---|---|---|---|---|---|---|---|---|---|---|---|
| | CPU Only | GPU Mode | Data Transfer | CPU Only | GPU Mode | Data Transfer | CPU Only | GPU Mode | Data Transfer | CPU Only | GPU Mode | Data Transfer |
| CRS | 0.13 | 3.29 | 0.83 | 0.55 | 1.29 | 0.8 | 0.63 | 1.21 | 0.74 | 1.1 | 1.37 | 0.91 |
| SMP-PCA | 1.21 | 3.16 | 1.5 | 1.65 | 1.46 | 0.73 | 1.66 | 3.53 | 1.49 | **4.5** | **3.69** | **1.73** |
| LTMM | **0.11** | **3.29** | 1.55 | **0.18** | **1.16** | 0.72 | 10.99 | 3.18 | 1.45 | 61.55 | 3.3 | 1.57 |

Table 6: The processing latency of combing AMM with GPU acceleration, the unit is ($\times 10^3 ms$). *Data Transfer* refers to the processing latency of data transferring between CPU and GPU under *GPU Mode*. Accuracy metrics $\epsilon$ is omitted, as it is the same as Table 4 and not changed by GPU utilization.

In our exploration of AMM's potential with hardware advancements, we experimented with pruning-based CRS and hybrid SMP-PCA algorithms, alongside the LTMM baseline, on a GPU-capable setup, i.e., one I7-13700K CPU and one RTX A6000 GPU. By switching LibTorch's backend from CPU to CUDA, without altering the *LibAMM* codebase, we analyzed the performance across four datasets: *ECO*, *UTM*, *RDB*, and *BUS*, as presented in Table 6.

**Our study shows that moving AMM algorithms to GPUs can speed up processing, but data transfer costs between CPU and GPU largely limit these benefits.** For instance, while LTMM benefits significantly from GPU acceleration, achieving up to a $95\%$ reduction in processing latency, AMM algorithms like SMP-PCA only manage an $18\%$ latency reduction. This discrepancy underscores the inefficiency in harnessing GPU computational power for AMM, largely due to the overhead from moving data. Notably, at larger scales, such as with the *BUS* dataset, SMP-PCA demonstrates a better fit for GPU acceleration than CRS, owing to its superior handling of memory operations. However, the overarching challenge remains the memory bottleneck, particularly the penalty from pending memory operations within the GPU and the substantial latency incurred during CPU-GPU data transfer—often accounting for about $50\%$ or more of the total processing time in most cases evaluated. To summarize, the main obstacle to fully leveraging GPU acceleration for AMM is not computational capacity but the efficiency of data movement, signaling an urgent call for innovative hardware-software co-designs that can effectively reduce memory overhead and optimize AMM performance.

## 3.3 Dataset Attributes

In this section, we explore how two key dataset attributes—*skewed numeric values* and *biased Non-zero (NNZ) regions*—affect AMM approximation accuracy ($\epsilon$). These attributes are common in real-world datasets and can introduce significant challenges for AMM algorithms. We use synthetic datasets to systematically isolate and evaluate their impact, with a focus on $\epsilon$ as latency variations among the algorithms were found to be negligible.

### 3.3.1 Skewed Numeric Values

*Skewed numeric values* refer to a distribution where numerical data is unevenly distributed, often resulting in a heavy tail, with a small number of values occurring frequently and dominating the dataset. To assess how skewed numeric values affect AMM algorithms, we designed experiments using two matrices of size $2500 \times 2500$. Matrix B was generated using a uniform random distribution (*torch::rand*), while matrix A was crafted to exhibit skewed numeric values by applying a Zipf distribution using *torch::pow* and *torch::multinomial*. We adjusted the skewness factor of the Zipf distribution from 1 to 2, gradually increasing the prevalence of a small set of dominant values. After this transformation, matrix A was normalized to match matrix B in the range $[0, 1]$.

> Observation 6. AMM exhibits stable accuracy unless large numeric value skewness.

Figure 4 shows the impact of skewed numeric values on $\epsilon$. **As the Zipf factor increases beyond 1.6, only a subset of algorithms (INT8, BLOCKLRA, CoOFD, VQ, and PQ) are able to maintain**

| Algorithms | | Processing Latency $l$ ($\times 10^2 ms$) | | | | | AMM Error $\epsilon$ | | | | | Approximation Impact Factor $\Delta E$ | | | | |
|---|---|---|---|---|---|---|---|---|---|---|---|---|---|---|---|---|
| | | PCA | MLT(500D) | MLT(2000D) | MLI | UT | PCA | MLT(500D) | MLT(2000D) | MLI | UT | PCA | MLT(500D) | MLT(2000D) | MLI | UT |
| Pruning-based | INT8 | 0.76 | 14.52 | 139.06 | 0.07 | 69.22 | 0.0541 | 0 | 0 | 0.03 | 0.1701 | 0 | 0.0011 | -0.0018 | 0.0001 | 0.3968 |
| | CRS | 0.55 | 94.32 | 100.43 | 0.01 | 10.93 | 0.4044 | 1.1099 | 1.8021 | 1.3933 | 0.0553 | 0.108 | 0.0483 | 0.0838 | 0.2558 | 0.0048 |
| | CS | 2.76 | 234.47 | 366.83 | 0.25 | 13.41 | 0.3876 | 0.8705 | 2.0562 | 1.3751 | 0.1046 | 0.0679 | 0.0577 | 0.0778 | 0.3008 | 0.0213 |
| Extraction-based | CoOFD | 4639.36 | 778.72 | 749.23 | 0.35 | 10361.91 | 0.2002 | 0.0683 | 0.1308 | 0.975 | 0.0079 | 0 | 0.0065 | 0.0091 | 0.6799 | 0.0191 |
| | BlockLRA | 9.67 | 2453.32 | 6281.38 | 4.62 | 355.13 | 0.524 | 0.0186 | 0.0283 | 0.9325 | 0.0031 | 0.1582 | 0.0009 | -0.0019 | 0.6346 | 0.0052 |
| | FastJLT | 86.16 | 50.72 | 2020.93 | 0.49 | 759.58 | 0.4408 | 0.9808 | 2.5515 | 1.6057 | 0.144 | 0.106 | 0.0608 | 0.0862 | 0.3603 | 0.2390 |
| | VQ | 12.27 | N.A. | N.A. | 33.56 | 1104.09 | 0.6341 | N.A. | N.A. | 0.9394 | 1.0024 | 0.3506 | N.A. | N.A. | 0.6869 | 1.0000 |
| | PQ | 61.99 | N.A. | N.A. | 2.96 | 2236.02 | 0.9989 | N.A. | N.A. | 1 | 1.0159 | 0.8671 | N.A. | N.A. | 0.6948 | 1.0000 |
| Hybrid | RIP | 8.8 | 19.87 | 98.43 | 0.04 | 21.86 | 0.4148 | 1.0116 | 1.9446 | 1.3913 | 0.0562 | 0.1186 | 0.0558 | 0.0596 | 0.319 | 0.0265 |
| | SMP-PCA | 6.77 | 44.62 | 369.23 | 0.07 | 31.84 | 0.4459 | 0.8641 | 1.9863 | 1.464 | 0.0307 | 0.0789 | 0.0589 | 0.0707 | 0.2007 | 0.0003 |
| | WeigthedCR | 2.51 | 23.62 | 133.13 | 0.14 | 17.78 | 0.4479 | 2.4964 | 5.2984 | 1.3469 | 0.0574 | 0.0963 | 0.0623 | 0.0335 | 0.2883 | 0.0041 |
| | TugOfWar | 78.88 | 284.42 | 736.13 | 0.39 | 404.3 | 0.3882 | 0.6275 | 1.6422 | 1.4119 | 0.1132 | 0.0752 | 0.0594 | 0.0914 | 0.3693 | 0.3535 |
| Baselines | NLMM | 16.8 | 6690.52 | 19744.48 | 14.74 | 14328.68 | 0 | 0 | 0 | 0 | 0 | 0 | 0.0011 | -0.0018 | 0 | 0 |
| | LTMM | 1.28 | 14.52 | 141.98 | 0.08 | 213.36 | 0 | 0 | 0 | 0 | 0 | 0 | 0 | 0 | 0 | 0 |

Table 7: Evaluation on applying AMM to downstream tasks. *MLT*, *MLI* and *UT* are abbreviations for *Machine Learning Training*, *Machine Learning Inference* and *Unitary Transformation*, respectively. VQ and PQ are excluded in MLT due to the $\geq 10^7 ms$ overhead of codebook rebuilding.

$\epsilon$ **below the critical 0.5 threshold.** For the remaining algorithms, error rates rise sharply, often exceeding 0.8, as skewness intensifies. This result highlights the pronounced sensitivity of most AMM algorithms to highly skewed datasets, with none exhibiting a boundary condition specifically designed to mitigate such effects.

### 3.3.2 Biased Non-zero (NNZ) Regions

*Biased Non-zero (NNZ) regions* describe matrices in which non-zero elements are disproportionately concentrated in specific areas, creating "active" regions, while large portions of the matrix remain sparse or filled with zeros. To explore the impact of biased NNZ regions, we generated a $2500 \times 2500$ matrix A initialized with zeros using *torch::zero*. We then introduced localized non-zero regions by populating the top-left sub-matrix with random values from *torch::rand*. By varying the percentage of rows and columns populated with non-zero elements from $10\%$ to $100\%$, we simulated varying degrees of bias in the NNZ regions.

> Observation 7. The biased NNZ regions influence AMM accuracy less significantly than skewed numeric distributions, with the notable exception of WEIGTHEDCR.

As shown in Figure 5, $\epsilon$ increases for all evaluated algorithms as NNZ bias intensifies. **However, the impact of NNZ bias is generally less severe than that of skewed numeric values.** For most algorithms, $\epsilon$ remained below 0.26, with the exception of WEIGTHEDCR, which showed significantly higher error rates due to its sampling-based approach and reliance on median value calculations, making it particularly vulnerable to localized NNZ concentrations.

### 3.4 Downstream Tasks

The outcomes of applying AMM to four downstream tasks are detailed in Table 7, where the approximation impact factor of AMM ($\Delta E$) is assessed through various application-specific errors. For example, reduced *PCA approximation quality* and increased *relative error of probability calculation* gauge the impact on PCA and *Unitary Transformation* tasks, respectively, and increased *classification error* for evaluating neural network performance in training and inference phases.

> Observation 8. Pruning-based and hybrid AMM outperform extraction-based AMM in diverse downstream tasks, highlighting the need for task-aligned approximation.

**Pruning-based and hybrid AMM strategies excel in reducing processing latency across various tasks, notably outperforming extraction-based methods in efficiency while maintaining acceptable error rates.** This is evident as INT8 and CRS significantly lower LTMM's processing latency, for instance, by up to 94% in unitary transformations, albeit with slight increases in error metrics. Hybrid approaches like SMP-PCA and RIP further demonstrate their capability to enhance processing speeds with larger matrices involved, such as in machine learning training and inference, showcasing a balanced trade-off between efficiency and error management. Conversely, extraction-based AMM generally lead to increased processing times and higher errors, highlighting a potential mismatch in their approximation strategies with the application requirements.

**AMM's numerical approximation focus can lead to semantic information loss, particularly in consecutive applications or nuanced interpretation tasks.** Extraction-based algorithms (BLOCKLRA, VQ, and PQ) exhibit higher $\epsilon$ in precision-centric tasks like PCA, where their

feature extraction does not align with the task's principal data characteristics, leading to elevated $\Delta E$. For example, these algorithms significantly increased $\Delta E$ in PCA due to mismatches in feature representation. The impact of AMM on machine learning training and inference highlights the importance of semantic preservation, which is often overlooked by current approximation strategies. In training, CRS's $\Delta E$ escalated from $0.05$ at a 500-D hidden layer to $0.08$ at 2000-D, indicating a greater risk of information loss with larger matrices. Similarly, in inference, extraction-based methods (CoOFD, BlockLRA, VQ, and PQ) failed to preserve semantic details, with CoOFD's $\Delta E$ reaching $0.68$. These findings underscore that AMM's numerical approximation focus can lead to semantic information loss, particularly in consecutive applications or nuanced interpretation tasks. However, SMP-PCA's resilience to error amplification in unitary transformations signals a potential for AMM strategies to achieve numerical and semantic fidelity.

## 4    Related Work

While AMM has been theoretically well-explored [21, 25, 13, 1, 9, 31, 29, 2, 22, 7, 17, 34, 36, 16, 6, 23], there exists a gap in benchmarks that evaluate these algorithms' performance in diverse real-world applications, particularly emphasizing accuracy alongside computational efficiency. Prior research has illustrated AMM's potential in specific tasks; however, these analyses often lack a holistic evaluation that considers various performance metrics [14, 34]. Our contribution aims to bridge this gap by offering an in-depth comparison of AMM algorithms across a range of practical tasks, thereby providing insights into their applicability and efficiency in real-world scenarios. Our approach sets a new precedent in the study of AMM by amalgamating an extensive array of algorithms and applications, moving beyond traditional theoretical assessments to explore their practical utility. By adopting a comprehensive benchmarking strategy, we illuminate the strengths and limitations of AMM, facilitating a richer understanding of its potential for enhancing computational processes. This endeavor not only aids in theoretical exploration but also enhances practical applications, marking a significant step forward in the evolution of AMM research.

## 5    Conclusion

This study sheds light on the impactful nuances of AMM algorithms within real-world applications. Our takeaways and inspirations for future works can be summarized as follows.

**Summary of Observations.** There are three major findings throughout this study. First, pruning-based (e.g., INT8, CRS) and hybrid (e.g., RIP, SMP-PCA) AMM outperform extraction-based AMM, especially in various downstream tasks (**O1, O2, O8**). Second, memory overhead is a common bottleneck, while some AMM (e.g., CRS, SMP-PCA) do succeed in optimizing memory access, the high cost of data transfer still limits border applications of AMM (**O4, O5**). Third, the tradeoff space between accuracy and efficiency is wide for some AMM (e.g., CRS, SMP-PCA) yet narrow for others (e.g., BlockLRA). It is further challenged by data distributions, and the existing error bound of AMM is not strong enough under severe skewness or bias of data distribution (**O3, O6, O7**).

**Impacts and Future Directions.** Our empirical insights offer a more comprehensive understanding of the strengths and limitations in current AMM to data science communities. Future work on AMM could further reduce the memory overhead while strengthening the error bound guarantees. We also envision a versatile and robust software-hardware co-design to better incorporate AMM with orthogonal optimizations, i.e., algorithmic design [32], parallel and distributed computing [26], and hardware technology [18], to better cater to the evolving demands of diverse computational landscapes.

## Acknowledgments and Disclosure of Funding

This project received partial support from the National Research Foundation, Singapore, and the Infocomm Media Development Authority under its Future Communications Research & Development Programme (FCP-SUTD-RG-2022-005), Ministry of Education AcRF Tier 2 grant (MOE-T2EP20122-0010, MOE-T2EP20221-0017), and a Nanyang Technological University startup grant (023452-00001). The perspectives, conclusions, or recommendations put forward in this material are exclusively those of the authors and do not mirror the views of the Ministry of Education, Singapore. Corresponding author is Shuhao Zhang.

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

# A  Details of Downstream Tasks

We discuss the details of four selected downstream tasks in the following. The application errors ($E$) when applying AMM or MM on those downstream tasks are summarized in Table 8.

| Algorithms | | PCA | MLT(500D) | MLT(2000D) | MLI | UT |
|---|---|---|---|---|---|---|
| Pruning-based | INT8 | **0.1323** | **0.0186** | **0.0166** | **0.2953** | **0.3968** |
| | CRS | **0.2403** | **0.0658** | **0.1022** | **0.5510** | **0.0048** |
| | CS | 0.2002 | 0.0752 | 0.0962 | 0.5960 | 0.0213 |
| Extraction-based | CoOFD | 0.1323 | 0.0240 | 0.0275 | 0.9751 | 0.0191 |
| | BlockLRA | 0.2905 | 0.0184 | 0.0165 | 0.9298 | 0.0052 |
| | FastJLT | 0.2383 | 0.0783 | 0.1046 | 0.6555 | 0.2390 |
| | VQ | 0.4829 | N.A. | N.A. | 0.9821 | 1.0000 |
| | PQ | 0.9994 | N.A. | N.A. | 0.9900 | 1.0000 |
| Hybrid | RIP | **0.2509** | **0.0733** | **0.0780** | **0.6142** | **0.0265** |
| | SMP-PCA | **0.2112** | **0.0764** | **0.0891** | **0.4959** | **0.0003** |
| | WeigthedCR | 0.2286 | 0.0798 | 0.0519 | 0.5835 | 0.0041 |
| | TugOfWar | 0.2075 | 0.0769 | 0.1098 | 0.6645 | 0.3535 |
| Baselines | NLMM | 0.1323 | 0.0186 | 0.0166 | 0.2952 | 0.0000 |
| | LTMM | **0.1323** | **0.0175** | **0.0184** | **0.2952** | **0.0000** |

Table 8: Application errors $E$ of AMM or MM. *MLT*, *MLI* and *UT* are abbreviations for *Machine Learning Training*, *Machine Learning Inference* and *Unitary Transformation*, respectively. VQ and PQ are excluded in MLT due to the $\geq 10^7 ms$ overhead of codebook rebuilding.

**Principal Component Analysis (PCA).** PCA is a popular statistical function for dimensionality reduction [33, 30]. It computes the rank-r approximation of a matrix $A$ as $\hat{A}_r$, and the application error $E$ is defined as $E = ||A - \hat{A}_r||/||A||$. PCA is required to compute the covariance matrix, and the involved MM can be replaced by AMM. We conducted the PCA task on the *SIFT10K* dataset following the methodology outlined in [34]. Because the number of rows is exceptionally small (128) in comparison to the substantial column count (10000), the tuning parameter $\omega$ is configured at 10% for PQ and VQ, as their adjustment is row-relevant (Section 2.3). For the remaining AMM, we set $\omega$ to 1%.

**Machine Learning Training.**  We implement the methodology outlined in prior work [1], incorporating AMM techniques into three fully connected layers of an MLP model during the machine learning training. We use different configurations of hidden layer dimensions in our evaluation, i.e., 500-D and 2000-D, which involve relatively smaller and larger weight matrices, respectively. There are thousands of Stochastic Gradient Descent (SGD) iterations in training, and each SGD iteration utilizes AMM to forward the training loss before randomly updating the weight matrices. We report the average $\epsilon$ of first 10 SGD iterations during training, as they are most meaningful for the training task. The $E$ is referred to as the *classification error* in validating neural networks [1]. We exclude VQ and PQ, because they necessitate costly rebuilding of the entire codebook from scratch for each new version of the weight matrix, and it requires $10^7$ ms $l$ for building codebook 1000 times even in 500-D hidden layer case.

**Machine Learning Inference.** For inference, we apply AMM to the final dense layer in the pre-trained model discussed in [6] and set $\omega$ to 2.56%. The machine learning model also works as classifiers in [6], and the meaning of $E$ is the same as that in the case of machine learning training. We employ *CIFAR100* datasets as the illustration example.

**Unitary Transformation.** Unitary transformation is one of the key operations in the scientific computing of quantum physics [15, 10], and it is also the building block of more sophisticated quantum transformations, such as the canonical transformation in Quantum Zero-Sum Games [20]. Specifically, it transforms the quantum state matrix $q$ into $Q$ by two consecutive MM with a unitary gate. All of these MM are possible to be replaced by AMM, which transforms $q$ into $\hat{Q}$ instead of $Q$. We report the average $\epsilon$ of these two multiplications. $Q^2$ is proportional to the probability of collapsing

into specific classic states when measured, and $E$ is hence formulated as $E = ||Q^2 - \hat{Q}^2||/||Q^2||$. For illustration purposes, we instantiate $q$ as one of the *QCD* matrices, let the unitary gate exhibit Gaussian distributions [5], and ignore the normalization constants.

