# OpenReview forum: "LibAMM: Empirical Insights into Approximate Computing for Accelerating Matrix Multiplication"
_NeurIPS.cc/2024/Datasets_and_Benchmarks_Track — NeurIPS 2024 Track Datasets and Benchmarks Poster_

### Official Review · Reviewer_pKWS · 2024-07-21
**Comprehensive Evaluation Framework for Approximate Matrix Multiplication Algorithms**

**Rating:** 7
**Confidence:** 3

**Review:**

**Evaluation of the Quality**

The quality of the paper is high. The authors have conducted a thorough evaluation of twelve AMM algorithms against two baselines using eight diverse datasets. The experiments are well-designed, with clearly defined configurations, datasets, tasks, and implementation details. The comprehensive nature of the evaluation ensures that the results are robust and reliable. Additionally, the paper provides a detailed discussion of the findings, making the analysis both insightful and valuable.

**Clarity**

The paper is well-structured with a clear motivation and problem formulation. It includes a concise literature summary of AMM, which sets the context effectively. The tables and figures are clearly labeled, enhancing the readability of the results. The experiment settings are meticulously described, ensuring that readers can replicate the study.

**Originality and Significance**

The methodology is original in its comprehensive evaluation of a broad set of AMM algorithms. While it builds on existing works in AMM, it offers a novel perspective by providing a standardized framework for evaluation and a detailed comparative analysis. The benchmark is significant for researchers and practitioners in the field of matrix multiplication, offering valuable insights into the trade-offs between different AMM strategies.

**Strengths:**

* Comprehensive evaluation of AMM algorithms
* Clear and well-structured presentation
* Robust experimental design
* Insightful analysis and discussion of findings

**Additional Feedback:**

* How would your framework handle new AMM algorithms that are not currently included?

**Clarity:**

The paper excels in clarity, with a logical flow of ideas and well-explained concepts. The figures and tables are effectively used to illustrate key points. The writing is precise and accessible. The documentation in the supplementary materials and appendices is also thorough

**Correctness:**

The experiments are well-designed, with clearly defined configurations, datasets, tasks, and implementation details. The results appear robust and reliable.

**Documentation:**

The documentation is comprehensive, offering clear descriptions of the experimental setup and evaluation metrics. The appendices and README in the repository provide additional context and detailed instructions for running and replicating the results.

**Ethics:**

No ethical concerns.

**Limitations:**

The assumptions and limitations are adequately described thoughtout the empirical studies.

**Opportunities For Improvement:**

The authors may improve instructions such that users can add new AMM algorithms to the framework.

**Relation To Prior Work:**

The paper includes a concise literature summary of AMM, effectively setting the context. It also highlights the shortcomings of current comparison methods and demonstrates how their work provides a comprehensive and impartial comparison.

**Summary And Contributions:**

This paper presents a framework for studying approximate matrix multiplication (AMM). The framework includes twelve AMM algorithms, two matrix multiplication baselines, eight benchmark datasets, and accompanying scripts. The paper provides a comprehensive comparison of the performance (processing latency, error, etc.) of these algorithms and highlights the key takeaways from their findings. The primary contributions of this work are the development of a standardized framework for AMM evaluation and the detailed comparative analysis of various AMM algorithms across multiple datasets.

---

> ### Author Rebuttal · Authors · 2024-08-16
>
> > The authors may improve instructions such that users can add new AMM algorithms to the framework.
>
> We appreciate the suggestion. While the current documentation provides a basic overview, we recognize the need for more detailed guidance. We will revise the documentation to include step-by-step instructions and examples, making it easier for users to integrate new AMM algorithms. This should encourage broader community engagement and expand the framework's capabilities.
>
> > How would your framework handle new AMM algorithms that are not currently included?
>
> Our framework is designed for modularity, allowing for the integration of new AMM algorithms. These can be added as separate modules within the existing codebase, ensuring seamless integration with the benchmarking tools. We’ll extend the documentation to include clear instructions on how to implement and benchmark new AMM methods, ensuring compatibility with the existing infrastructure. This flexibility is intended to foster community-driven development and broaden the framework's applicability.

---

> ### Comment · Reviewer_pKWS · 2024-08-28
>
> Thank you for your response. I am keeping my score as is.

---

### Official Review · Reviewer_izvp · 2024-07-24
**Extensive practical evaluation framework for approximate matrix multiply**

**Rating:** 7
**Confidence:** 3

**Review:**

The authors motivate the necessity of the current study and its significance for the research community: while there exists a large number of AMM methods and studies of their theoretical properties, often with contradictory guidelines for algorithm selection, there is little focus on a fair comparison of their performance in extensive practical setups.

Pros:
* the paper is well written and well motivated and the experiments are followed by detailed discussions resulting in many interesting key observations
* the algorithm selection, datasets and practical methods used for evaluation are clearly explained
* the evaluation is extensive, accounting for many dimensions, such as datasets properties (skewness and biased non-zero regions), practical methods (PCA, machine learning training and inference, Unitary Transformations) as well as memory and compute cycles and operations and their impact on the overall latency and errors.

Cons:
* the current work studies matrices with min-max normalization between -1 and 1, while using other types of normalisations and value ranges might provide different results for some of the methods
* the impact of scale is observed only for the particular case of "tall" matrices (with a large number of rows)
* as noted by the authors, some of the methods might benefit from algorithm-specific optimizations that are excluded due to their lack of generality

**Strengths:**

* The necessity of a comprehensive evaluation of AMM through the practical lens is well motivated
* the study is includes a reasonably sized selection of 12 algorithms and several datasets, as well as synthetic matrices of adjustable shapes, sensitivity to dataset characteristics as well as out-of-the-box (without hardware-specific optimizations) potential in a parallelized setup as well as detailed analysis on dimensions of the latency to approximate error trade-offs
* use of static compilation for consistency across experiments
* the provided code is well-structured and easy to explore

**Additional Feedback:**

Figures 1 and 2 are difficult to follow due to the large number of methods that are evaluated. It seems that a lot of methods show robust latency within large intervals of the the tuning knob value range (for example FastJLT and TugOfWar), as opposed to error which has a much more pronounced variance. I believe an extra visualization of the latency and error ratio for each method would be really helpful.

**Clarity:**

Yes, the paper is well written, well structured and the the experiments and their interpretations are clear.

**Correctness:**

Yes, the evaluation methods and experiment design seem to be appropriate and performed correctly. The authors focus on AMM latency and errors, as well as their trade-off axes, but also on divergence from MM in actual practical applications, such as machine learning training and inference, PCA and Unitary Transformations. The evaluation is conducted with parallel observations towards dataset properties, matrix scales and tuning knob factor.

**Documentation:**

The authors utilize several publicly available datasets, as well as synthetic random matrices.

**Ethics:**

There are no ethical concerns with the submission.

**Limitations:**

There are mentions of limitations of the current study, such as more advanced normalization techniques and dataset-specific optimizations not being taken into account.

**Opportunities For Improvement:**

* The effect of data scale is observed by adjusting the number of rows in A, resulting in "tall" matrices. Could some of the algorithms present different scalability properties for the width dimension of matrices (number of columns is varied) that might potentially result in different trends and observations? These types of matrices might also be common in real world applications.
* As it was mentioned in the paper, other types of normalization techniques (and value ranges other than -1 to 1), as well as algorithm-specific optimizations are missing in the current evaluation. I hope these will be pursued as future work.

**Relation To Prior Work:**

Yes, the authors' premise is that AMM methods have been well studied theoretically and practically in specific setups and propose the LibAMM as an unified, more general and extensive practical evaluation framework.

**Summary And Contributions:**

The authors conduct a comprehensive study of 12 approximate matrix multiplication (AMM) methods and their capabilities and limitations in a practical setup, beyond their more well-studied theoretical properties. The AMM methods are categorised in pruning-based, extraction-based and hybrid methods and evaluated in the context of 8 real-world datasets and 4 statistics and machine learning applications, with a focus on approximation fidelity, downstream application error and latency/memory efficiency and their trade-off dimensions. The baselines are the naive nested loop implementation of matrix multiplication and LibTorch optimized implementation.

Some of the key observations include:
* pruning-based and hybrid methods seem to outperform extraction-based methods in terms of latency. In contrast, pruning-based methods also provide higher errors, as well as poorer scaling with data sizes compared to extraction-based methods. Hybrid methods offer a trade-off between efficiency and accuracy, as well as better scaling with data size than pruning-based methods.
* the latency / error trade-off has a complex landscape through the tuning knob $\omega$
* memory stalls are a critical bottleneck across all AMM algorithms, while computing stalls seems to be problematic for certain algorithms, but pruning-based and hybrid AMM are more successful in mitigating memory stall growth with data size, outperforming the LibTorch optimized implementation for certain matrix sizes
* downstream applications results further support previous findings, with pruning-based and hybrid methods outperforming extraction-based ones

---

> ### Author Rebuttal · Authors · 2024-08-16
>
> > The current work studies matrices with min-max normalization between -1 and 1, while using other types of normalizations and value ranges might provide different results for some of the methods. As it was mentioned in the paper, other types of normalization techniques (and value ranges other than -1 to 1), as well as algorithm-specific optimizations, are missing in the current evaluation. I hope these will be pursued as future work.
>
> Our work normalizes matrices between -1 and 1 for consistency, but we recognize that different normalization techniques or value ranges could yield different results for some methods. We'll clarify this limitation in the manuscript and consider broader normalization techniques in future studies.
>
> > The impact of scale is observed only for the particular case of "tall" matrices (with a large number of rows). Could some of the algorithms present different scalability properties for the width dimension of matrices (number of columns is varied) that might potentially result in different trends and observations? These types of matrices might also be common in real-world applications.
>
> The impact of scale was examined by varying the number of rows, creating 'tall' matrices. We acknowledge that varying the number of columns to create 'wide' matrices could reveal different scalability properties and trends. This limitation will be noted, and we suggest future work explore scalability in both dimensions.
>
> > As noted by the authors, some of the methods might benefit from algorithm-specific optimizations that are excluded due to their lack of generality.
>
> Certain methods could benefit from algorithm-specific optimizations, which we excluded to ensure a fair comparison. We’ll clarify this decision and suggest that future work include these optimizations with appropriate disclaimers.
>
> > Figures 1 and 2 are difficult to follow due to the large number of methods that are evaluated. It seems that a lot of methods show robust latency within large intervals of the tuning knob value range (for example FastJLT and TugOfWar), as opposed to error which has a much more pronounced variance. I believe an extra visualization of the latency and error ratio for each method would be really helpful.
>
> Figures 1 and 2 are challenging to follow due to the large number of methods. We agree that some methods show robust latency over a broad range of tuning knob values, while error varies more. We’ll add a visualization of the latency-to-error ratio for each method to make the results easier to interpret.

---

> > ### Comment · Reviewer_izvp · 2024-08-28
> >
> > Thank you for the response! I have increased the rating following a careful look at all the reviewer discussions.

---

### Official Review · Reviewer_zqbq · 2024-08-02
**Benchmark to evaluate speed-up of matrix multiplication through Approximate Matrix Multiplication methods**

**Rating:** 7
**Confidence:** 3

**Review:**

- __Originality__: The typology of the benchmark is classic in the HPC community and may benefit from better reviews from experts in this area of research. In the conclusion of the paper, the claim about memory overhead being a major finding is more of a sanity check as in high-performance computing it is well known that memory accesses (for different memory levels: cache, RAM, disk...etc) constitute a bottleneck as they are order of magnitude slower than computations.
 - __Quality__: The proposed benchmark and analysis are good quality.
- __Clarity__: The paper is very dense and the results presentation could be improved (tables are hard to read, bold numbers are hard to understand, color code of plots does not group methods of the same class).
- __Significance__: Optimisation of matrix multiplication can have a significant impact across the scientific community and also for industry.

**Strengths:**

- The benchmark includes a variety of methods from the literature, a variety of datasets, and downstream tasks.
- The analysis extracts generalizable findings (also a weakness as optimizations are key in HPC).
- The code is publicly available.

**Additional Feedback:**

## Open question

Due to the numerical precision of float32 representation, even MM has an error > 0. Would it be interesting to maybe use as a baseline a representation with higher fidelity, let's say float64, and compare all methods to the results of this baseline (trying to be closer to what could be called "the truth")?

**Clarity:**

- Sec. 2.3.: "The tuning parameter ω plays a crucial role in managing the trade-off between computational efficiency (l) and accuracy (ε)in AMM algorithms." So why are we not doing a multi-objective analysis (Pareto-fronts)?
- Sec. 2.3.: It would be great to explicit the support of the $\omega$ parameter and if this support is similar across AMM methods.
- Table 3: "Latency proportion" could be clarified in the legend.
- The paper mentions in the intro that it does not focus on hardware-specific optimization. However in Sec. 3./Implementation, it mentions the use of AVX-512 that is specific to x86 architecture according to my understanding. Could you clarify which type of hardware-specific you do not wish to consider because they are not generalizable (basically what would or would not be a generalizable optimization)?
- I am surprised to not see error bars in the experimental results (e.g., Table 4) and no mention of the number of repeated tests for each setting (I guess latency measures are noisy?).
- I don't understand the "bold" font rule in Table 4.
- Figure 1/2: too small, presentation would be better with larger figures (even if we can zoom in).
- Figure 1/2: would be helpful to have a style for plotting that is coherent with method classes (pruning, extraction, hybrid, baselines) to see more easily group trends.

**Correctness:**

- Assuming the limited horizon (in terms of hardware, system) of the conclusion, the methodology does not seem flawed to me.

**Documentation:**

The repo is fairly documented but it could be improved.

**Limitations:**

- The benchmark includes a limited environment (hardware and OS). There is only 1 CPU type, 1 GPU type and many things in the environment can be mixed. Such a benchmark would be beneficial if it could be easily downloaded and setup on different systems to collect experimental data in different contexts and expand the analysis.
- Limitations of the work are not clearly stated in the paper.

**Opportunities For Improvement:**

- Introduction: Takeaways 1 and 4 are hard to differentiate maybe rephrase. They both put in bold "pruning and hybrid is better than extraction".
- Presentation of the README in the repo could be improved, especially "How to XXX" sections.
- Sec. 2.1: "by measuring the time from $A$ and $B$ are presented to $\tilde{C}$  is eventually produced" rephrase.
- Sec. 2.1: "frobenius normalized accuracy" maybe calling it an "error" would be more appropriate as generally we maximize accuracy and minimize the error.
- Sec. 2.1: maybe introduce the $E_{XXX}$ before $\Delta E$, if I understand correctly this corresponds to the quality metrics of the downstream task.
- Table 1: typo "Purning" -> "Pruning" x4

**Relation To Prior Work:**

The paper introduces methods from the literature (pruning-based, extraction-based, and hybrid). The paper situate limitations of existing studies: (1) task (e.g., training and inference), (2) dataset diversity (e.g., synthetic, "realistic"), (3) implementation (e.g., specificities of the implementation such as (JIT) compilation or loops), (4) over pessimistic baseline (poorly optimized and not reflecting real implementation used).

**Summary And Contributions:**

The paper is about proposing a benchmark for approximate matrix multiplications (multiple algorithms and datasets for experimentation). The paper focused on providing (1) a performance analysis of MM and AMM algorithms that focuses on (principles and generally applicable concepts (does not focus on system software/hardware specific optimizations), (2) eight "realistic" datasets and (3) 4 applications.

---

> ### Author Rebuttal · Authors · 2024-08-16
>
> > The benchmark includes a limited environment (hardware and OS).
>
> The benchmark is streamlined for reproducibility by limiting the hardware and OS, minimizing variability that could obscure algorithm performance. Only one CPU and GPU type are included to ensure results reflect the tested methods rather than hardware differences. The benchmark is modular, making it easy for users to adapt to other configurations, broadening its applicability. We are also preparing Docker containers for easier adoption.
>
> > Limitations of the work are not clearly stated in the paper.
>
> We discuss limitations in the methodology section but agree they should be more prominent. We'll add a dedicated section to clearly outline these constraints.
>
> > Sec. 2.3.: "The tuning parameter ω plays a crucial role in managing the trade-off between computational efficiency (l) and accuracy (ε) in AMM algorithms." So why are we not doing a multi-objective analysis (Pareto-fronts)?
>
> While ω is key to balancing efficiency and accuracy, our focus was on providing a detailed empirical analysis. A multi-objective analysis would add complexity and could obscure direct comparisons. We prioritized clarity but acknowledge the value of Pareto-front analysis for future work.
>
> > Sec. 2.3.: It would be great to explicit the support of the parameter and if this support is similar across AMM methods.
>
> We will revise Section 2.3 to detail the range of ω values for each AMM method and clarify if this range is consistent across algorithms.
>
> > Table 3: "Latency proportion" could be clarified in the legend.
>
> We’ll clarify in Table 3’s legend that 'Latency proportion' refers to the percentage of total processing time spent on AMM-replaceable matrix multiplication operations in the given tasks.
>
> > The paper mentions in the intro that it does not focus on hardware-specific optimization. However, in Sec. 3./Implementation, it mentions the use of AVX-512 that is specific to x86 architecture. Could you clarify which type of hardware-specific you do not wish to consider because they are not generalizable?
>
> We avoided optimizations requiring custom modifications or tightly coupled to specific hardware setups. AVX-512, used in our work, is a standard feature in the x86 ecosystem, ensuring generalizability. We avoided manual AVX-512 implementations like assembly tricks, relying instead on LibTorch functions. We’ll clarify these points in the manuscript.
>
> > I am surprised to not see error bars in the experimental results (e.g., Table 4) and no mention of the number of repeated tests for each setting (I guess latency measures are noisy?).
>
> Please refer to the attached PDF for the error bar in running QCD data over 20 repeated tests (Following Table 4 settings). As expected, the relative error of pruning-based AMM is the largest but still no more than +-6%, which is due to the inherited randomness. On the contrary, hybrid AMM involves smaller relative error, and the relative error bar in extraction-based AMM is even less than that of LTMM. This is because their feature extraction is relatively deterministic.
>
> > I don't understand the "bold" font rule in Table 4.
>
> Bold font in Table 4 highlights the best performance in latency or error for each dataset. We’ll clarify this in the table’s legend.
>
> > Figure 1/2: too small, presentation would be better with larger figures (even if we can zoom in).
>
> We’ll resize Figures 1 and 2 to be larger for better readability.
>
> > Figure 1/2: would be helpful to have a style for plotting that is coherent with method classes (pruning, extraction, hybrid, baselines) to see more easily group trends.
>
> We’ll update Figures 1 and 2 with a coherent style, using distinct markers for each method class to highlight group trends.
>
> > Due to the numerical precision of float32 representation, even MM has an error > 0. Would it be interesting to maybe use as a baseline a representation with higher fidelity, let's say float64, and compare all methods to the results of this baseline (trying to be closer to what could be called "the truth")?
>
> Using float64 as a baseline is a valuable idea. This would provide a more precise reference point, allowing a better assessment of approximation errors across methods. We’ll consider this approach for future work.

---

> ### Comment · Reviewer_zqbq · 2024-08-16
>
> The answers from the authors satisfied my requests. I raised my score.

---

### Author Rebuttal · Authors · 2024-08-30

> (To all reviewers) Codebase improvement

We thank all the reviewers for the careful examination of our codebase. According to the suggestions, we have already made the following changes at https://github.com/intellistream/LibAMM.git by August 30th.
1. added a refman.pdf (doxygen-based, auto-generated) for a more detailed description of the code. A quick glance at algorithm details is documented in Section 7.4 of refman.pdf.
2. added one-key build scripts (as illustrated in README.md)
3. revise our Python API and enable pybind & pip, we have also provided a tutorial for our new version of Python API (benchmark/scripts/PyAMM/tutorial.ipynb). Now it looks more friendly to pytorch users.

---

### Decision · Program_Chairs · 2024-09-26

**Decision:**

Accept (Poster)

**Comment:**

The paper proposes a benchmark for approximate matrix multiplications (multiple algorithms and datasets for experimentation), focusing on (1) a performance analysis, (2) eight datasets and (3) four applications.  All reviewers agreed that the paper made a useful contribution.